# In Vitro Evaluation of Five Newly Isolated Bacteriophages against *E. faecalis* Biofilm for Their Potential Use against Post-Treatment Apical Periodontitis

**DOI:** 10.3390/pharmaceutics14091779

**Published:** 2022-08-25

**Authors:** Marie Voit, Andrej Trampuz, Mercedes Gonzalez Moreno

**Affiliations:** 1Charité-Universitätsmedizin Berlin, Corporate Member of Freie Universität Berlin, Humboldt-Universität zu Berlin, Centre for Musculoskeletal Surgery, Augustenburger Platz 1, 13353 Berlin, Germany; 2Berlin Institute of Health at Charité-Universitätsmedizin Berlin, BIH Center for Regenerative Therapies (BCRT), Charitéplatz 1, 10117 Berlin, Germany

**Keywords:** *Enterococcus faecalis*, biofilm, persistent root canal infection, bacteriophage isolation, bacteriophage-antibiotic combination, isothermal microcalorimetry, apical periodontitis

## Abstract

State-of-the-art treatment of root canal infection includes the use of mechanical debridement and chemical agents. This disinfection method is limited, and microorganisms can remain in the canal system. *Enterococcus faecalis* appears with a high prevalence in secondary and persistent root canal infections and can be linked to endodontic treatment failure due to its various resistance mechanisms. Here, we evaluated the activity of newly isolated bacteriophages against clinical isolates of *E. faecalis* (including one vancomycin- and gentamicin-resistant strain) as a single treatment or in combination with gentamicin and vancomycin. For the resistant strain, daptomycin and fosfomycin were tested. Sixteen *E. faecalis* strains were used to screen for the presence of bacteriophages in sewage. Five different bacteriophages were characterized in terms of virion morphology, host range and killing-kinetics against each *E. faecalis* host strain. To investigate the antibiofilm effect of antibiotic and phages, *E. faecalis* biofilm was grown on porous glass beads and treated with different antibiotic concentrations and with isolated bacteriophages alone or in staggered combinations. A strong biofilm reduction was observed when phages were combined with antibiotic, where combinations with gentamicin showed a better outcome compared to vancomycin. Regarding the resistant strain, daptomycin had a superior antibiofilm effect than fosfomycin.

## 1. Introduction

Apical periodontitis is a biofilm-related infectious disease, which leads to an inflammatory reaction of the periapical tissue followed by destruction of the bone [1,2]. Post-treatment apical periodontitis counts as the main reason for endodontic treatment failure [3] and is categorized either as reinfection (acquired or emergent), persistent, or recurrent infection in teeth with a previous root canal treatment [4]. The prevalence of this disease varies widely and differs globally [5]. Previous studies in Austria and Germany have shown a prevalence of up to 61% [6,7].

The common method of root canal treatment includes a combination of mechanical debridement with nickel-titanium files and chemical disinfection with irrigants such as sodium hypochlorite and EDTA with the purpose to remove microorganisms, debris, and necrotic pulp tissue, followed by root canal obturation and a coronal restauration [8,9]. Nevertheless, this treatment approach is often linked to certain deficiencies. Due to lateral canals, apical ramifications, dentinal tubules and isthmuses, pathogens remain in the canal system and can cause a remnant inflammation [10,11,12]. Moreover, the shape of root canals plays an important role when it comes to mechanical debridement. For instance, in oval canals, 20–40% of the canal wall stays untouched [13,14].

*Enterococcus faecalis* is linked to post treatment root canal infections with a high prevalence up to 77% [15,16] due to its resistance against high pH and common endodontic irrigants. Additionally, *E. faecalis* can form a biofilm formation in the root canal, being also able to penetrate dentin and adhere to collagen [17,18]. Inability to eradicate *E. faecalis* in the canal system may lead to persistent infection and loss of tooth structure.

*E. faecalis* is a Gram-positive facultative anaerobe bacterium that occurs in the human gastrointestinal tract but can also become an opportunistic pathogen that can lead to life threatening systemic infections such as endocarditis, meningitis, urinary tract infections, and can be linked to periprosthetic joint infections [19,20,21]. Due to the acquisition of antibiotic resistance, resulting in multidrug-resistant strains such as vancomycin-resistant enterococci [22,23], persistent root canal infections of *E. faecalis* origin become clinically relevant.

As previously shown [24,25], bacteriophage therapy appears as a promising alternative to target *E. faecalis* and has barely been explored in dentistry. Bacteriophages (also referred as phages) are viruses that specifically infect bacteria. The emergence of multidrug-resistant bacterial infections has led to recent efforts to investigate and promote phage therapy to treat a multitude of infections [26]. Thus, in this study, we evaluated the antimicrobial and antibiofilm activity of five newly isolated phages targeting clinical *E. faecalis* isolates alone and in combination with antibiotics in view of their potential local application in the control of intraradicular root canal infections.

## 2. Materials and Methods

### 2.1. Bacterial Strains and Antimicrobial Agents

Two *E. faecalis* ATCC strains (19,433, 29,212) and 14 clinical isolates (EF 03 to EF 16) from patients with biofilm-related infections, including one vancomycin- and gentamicin-resistant strain (EF 04), were used in this study. Bacteria were stored in 25% glycerol solution at −20 °C. All bacterial strains were cultured on Tryptic Soy Broth (TSB) (US Biological, Eching, Germany) or Müller Hinton Broth (MHB) (Roth, Karlsruhe, Germany) for liquid cultures or in TSB with addition of 15 g/L (TSA) or 6 g/L (Top-Agar) agar (Sigma-Aldrich, Steinheim, Germany).

Gentamicin injectable solution (Ratiopharm, Ulm, Germany), vancomycin powder (Hexal, Holzkirchen, Germany), daptomycin powder (Novartis, Basel, Switzerland), and fosfomycin powder (InfectoPharm, Heppenheim, Germany) were purchased from the respective manufacturers. Powdered antibiotics were reconstituted using sterile pyrogen-free water. MHB media for testing daptomycin and fosfomycin was supplemented with 50 mg/L Calcium chloride and 25 mg/L Glucose 6-phosphate, respectively.

### 2.2. Bacteriophage Isolation

Sewage samples from wastewater purification plants and cesspools in Berlin were used to screen for the presence of lytic phages. All samples were enriched with the 16 *E. faecalis* strains from the collection. After incubation for 24 h at 37 °C and 150 rpm, all samples were centrifuged (20 min, 7830 rpm, 4 °C) and filter-sterilized using a 0.22 µm filter. The enriched wastewater was then spotted on the bacterial lawn of each *E. faecalis* strain and incubated overnight at 37 °C. Afterwards, each plate was screened for the presence of clear lysis zones and single plaques were further isolated. Then, single phage solutions were produced via propagation from a single plaque in solid culture. Briefly, a single phage plaque was picked with a sterile toothpick and was then sticked up to 20 times into an agar plate with a bacterial lawn of the host strain. Afterwards, a sterile paper stripe was used to swipe over the punctures to spread the phages all over the agar plate and was incubated overnight at 37 °C. After incubation, 5 mL of saline magnesium (SM) buffer (10 mM Tris-HCL, pH 7.8, 1 mM MgSO_4_) were added to the agar plate and incubated at 4 °C for 24 h. Then, the SM buffer with the eluted phages was collected, centrifuged (20 min, 7830 rpm, 4 °C) and filter-sterilized by using a 0.22 µm filter. Titration was performed to determine the phage titer on the host strain and the phage stocks were stored at 4 °C. Isolated phages were named as “CUB” followed by a number corresponding to the *E. faecalis* host strain where they were isolated and propagated (e.g., phage CUB_EF03 was isolated and propagated using the *E. faecalis* strain EF 03). Each phage was tested using its own host bacterial strain (which generally corresponded to the strain where a larger and clearer plaque morphology and/or the presence of a halo (data not shown) was observed) for the analysis of antibiofilm activity.

### 2.3. Morphological Analysis by Transmission Electron Microscopy

The virion morphology characteristics were visualized by transmission electron microscopy (TEM) using the negative staining technique. A volume of 15 µL of phage solution was dropped onto parafilm prior transferal onto a Ni-mesh grid (G2430N; Plano GmbH, Wetzlar, Germany) that has previously been carbon-coated and glow discharged (Leica MED 020, Leica Microsystems, Wetzlar, Germany). The grid is then let to adsorb for 10–15 min at room temperature and then washed three times with Aquadest and treated with 1% aqueous uranyl acetate (SERVA Electrophoresis GmbH, Heidelberg, Germany) for 20 s for negative staining. Afterwards, excess staining was removed with filter paper. Grids were air dried and then imaged by TEM using a Zeiss EM 906 microscope (Carl Zeiss Microscopy Deutschland, Oberkochen, Germany) at a voltage of 80 kV. The image processing program ImageJ [27] was used for phage size measurements.

### 2.4. Host Range Analysis

To evaluate the spectrum of *E. faecalis* strains that can be infected by the newly isolated phages, a host range was determined by Top-Agar overlay spot assays. Bacterial overnight cultures (14 clinical strains and 2 ATCC strains) were prepared in TSB. One hundred microliters of the bacterial culture was added to 4 mL Top-Agar and the mixture was poured onto a TSA plate and allowed to dry for a few minutes at room temperature. Ten-fold serial dilutions of each phage were prepared in SM buffer and 10 µL of each dilution were spotted on the bacterial lawn and incubated at 37 °C overnight. After incubation, the plates were checked for phage plaque formation, indicative of the antimicrobial activity.

### 2.5. Biofilm Time-Killing Assay

A time-killing curve assay to quantify the antibiofilm effect of the siphovirus-like phages (CUB_EF03, CUB_EF06, CUB_EF10, CUB_EF14) was performed as previously described [28]. Shortly, bacterial biofilms were formed on sterile 4 mm sintered porous glass beads (ROBU, Hattert, Germany) for each *E. faecalis* host strain. To this end, beads were placed into a 24-well-plate (Corning Inc., Corning, NY, USA) and covered with 1 mL TSB inoculated with 1:100 dilution of an overnight culture. Beads were then incubated at 37 °C and 150 rpm for 24 h in humid conditions. Thereafter, beads were dip-washed with sterile 0.9% saline to remove non-adhered planktonic cells and transferred into microcentrifuge tubes containing 1 mL of fresh TSB broth inoculated with 10^8^ PFU/mL phage. Samples were incubated at 37 °C for 0 h, 2 h, 4 h, 6 h and 24 h. Then, biofilm-embedded cells were recovered by sonicating the tubes for 10 min in a BactoSonic ultrasound bath at 40 kHz and 0.2 W/cm^2^ (BANDELIN electronic, Berlin, Germany). The bead was removed from the tube and the sonication fluid was centrifuged for 1 min at 16,000× *g* and 4 °C. The supernatant was discarded and the bacterial pellet resuspended in 1 mL phosphate-buffered saline. The centrifugation and resuspension step was carried out four times to wash out remaining phage particles from the samples. Finally, ten-fold serial dilutions of the resuspended fluid were plated on TSA and after 18–24 h incubation at 37 °C, recovered biofilm cells were quantified by colony counting. The bacterial concentration was calculated with the Equation (1):(1)$$\mathrm{Bacterial}\;\mathrm{Count}\;\left(\mathrm{CFU}/\mathrm{mL}\right)=\frac{\mathrm{No}.\;\mathrm{of}\;\mathrm{colonies}\;\mathrm{counted}\;\times \;\mathrm{dilution}\;\mathrm{factor}}{\mathrm{volume}\;\mathrm{in}\;\mathrm{mL}\;\mathrm{spread}\;\mathrm{on}\;\mathrm{culture}\;\mathrm{plate}}$$

Two biological replicates were carried out with technical duplicates. Data was expressed as mean ± standard deviation (SD) and plotted as bacterial count (CFU/mL) over time (h) using GraphPad Prism 6 software (GraphPad Software, La Jolla, CA, USA).

### 2.6. Phage Antimicrobial Activity Evaluated by Isothermal Microcalorimetry

The antimicrobial effect of the selected phages against biofilms was also monitored by isothermal microcalorimetry (IMC) as in a previous study [29] with some modifications. *E. faecalis* biofilm was grown on porous glass beads as previously described and single glass beads were introduced into sterile glass ampules containing 500 µL of TSB and 500 µL of phage solution (10^8^ PFU/mL). Ampoules where then inserted into the isothermal calorimeter TAM III (TA Instruments, New Castle, DE, USA) and the heat production was measured over 48 h. This experiment was performed with biological triplicates. Plots of heat (J) over time (h) were prepared using RStudio [30].

### 2.7. Antibiotic Susceptibility Determination

The broth microdilution method was used to determine the minimum inhibitory concentration (MIC) for each antibiotic in MHB. An approximately 5 × 10^5^ CFU/mL bacterial inoculum was prepared and exposed to two-fold serial dilutions of each antibiotic during 24 h at 37 °C. The lowest concentration of antibiotic that completely inhibited visible growth (clear wells) was defined as the MIC. The Clinical and Laboratory Standards Institute (CLSI) guidelines [31] were followed for the interpretation of susceptibilities. Consequently, *E. faecalis* strains were considered susceptible to vancomycin when MIC ≤ 4 µg/mL, to fosfomycin when MIC ≤ 64 µg/mL and to daptomycin when MIC ≤ 2 µg/mL. Susceptibility testing on *E. faecalis* strains is not usually recommended for gentamicin, although it can be used to screen for high-level aminoglycoside resistance (MIC > 128 µg/mL) [32].

### 2.8. Phage-Antibiotic Combinations against Biofilm

First, 24 hour-old-biofilms were formed on porous glass beads as described above, deep washed with 0.9% NaCl and exposed to either phage (10^8^ PFU/mL), antibiotic at 1× MIC, 10× MIC and 100× MIC concentrations, or the combination of both phage and antibiotic, in a final volume of 1 mL and incubated at 37 °C for 24 h. In addition, a paired combination of two antibiotics (at 1× MIC and 10× MIC) and the phage was also included to evaluate potential antibiotic interactions. For all phage/antibiotic combinations, biofilms were exposed first to phages for 4 h at 37 °C after which the antibiotic was added and incubated for further 20 h at 37 °C. After a total incubation of 24 h, treated biofilm-beads were rinsed three times with 0.9% NaCl, placed in sterile glass ampules with 1 mL fresh TSB and inserted in the calorimeter, where heat produced by viable bacteria present in the bead after 24 h of treatment or no treatment (growth control) was monitored for 48 h. Experiments with each *E. faecalis* strain were conducted using two biological replicates, each with technical duplicates.

### 2.9. Assessment of Bacterial Resistance Development to Phage

The development of resistance of *E. faecalis* strains EF 10 and EF 14 to phage was assessed by Top-Agar overlay spot assays. To this end, the supernatant of EF 10 and EF 14 biofilm samples co-incubated with phage CUB_EF10 or CUB_EF14, respectively, were collected after the 48-hour calorimetric assay and plated on an agar plate. After 24 h incubation of the agar plates at 37 °C, a single bacterial colony (referred as colony-variant) was collected for the resistance analysis. The susceptibility of each retrieved bacterial colony-variant to all siphovirus-like phages (CUB_EF03, CUB_EF06, CUB_EF10 and CUB_EF14) was assessed by Top-Agar overlay spot assay. Bacterial colony-variants were considered resistant to the phage when no lysis zone could be observed on the plate.

## 3. Results

### 3.1. Phage Isolation and TEM Visualization

Five distinct phages were isolated on clinical *E. faecalis* strains from sewage and the phage morphologies were examined by TEM (Figure 1). Four phages (CUB_EF03, CUB_EF06, CUB_EF10 and CUB_EF14) displayed an eicosahedral head and a non-contractile tail, with no visible tail fibers, but a tail tip, corresponding to a siphovirus morphology, whereas phage CUB_EF04 displayed a podovirus morphology with an eicoshedral head (45.18 nm in diameter) and a short, noncontractile tail (21.73 nm in length).

CUB_EF03 had a head diameter of 54.83 nm, a tail tube of 188.15 nm in length and 8.91 nm in width and a visible tail tip of 32.81 nm long. Similarly, CUB_EF06 presented a head size of 54.48 nm in diameter and a tail tube of 195.76 nm in length and 7.79 nm in width. CUB_EF10 displayed the biggest head size with a diameter of 64.12 nm, a tail tube 189.72 nm long and 11 nm wide and a visible tail pin of 22.6 nm. Finally, CUB_EF14 had a head size of 59.45 nm in diameter, a long tail tube of 200.56 nm and 9.37 nm wide, and a tail tip of 16.31 nm.

### 3.2. Host Range

According to Figure 2, phage CUB_EF14 showed the broadest host range, active against 7 out of 16 (44%) tested *E. faecalis* strains, whereas phage CUB_EF10 displayed the narrowest host range, only active against the host strain (EF 10) used during the isolation step. Five bacterial strains (31%) did not show susceptibility to any of the tested phages.

### 3.3. Time-Killing Assay

The killing kinetics revealed a significant reduction of *E. faecalis* biofilm cells after 4 to 6 h of co-incubation with phage (Figure 3). The strongest antibiofilm activity was observed with CUB_EF10, which showed a CFU reduction of more that 3-log (99.9%) against EF 10 after 4 h co-incubation (Figure 3C). Nevertheless, the treated biofilm of all four tested strains showed an increase in CFU counts at 24 h, with values comparable to those observed for the untreated samples, possibly indicative of the emergence of bacterial resistance to phage.

### 3.4. Phage-Antibiotic Combinations against E. faecalis Biofilms

Prior determination of the combined effect of phage and antibiotic against biofilm, the MIC values for the selected antibiotics were determined on each *E. faecalis* strains (Table 1).

All four tested *E. faecalis* strains were susceptible to vancomycin and showed an absence of high-level of aminoglycoside resistance when tested with gentamicin. As for the vancomycin and gentamicin resistant strain EF 04, it was susceptible to fosfomycin and daptomycin.

The antibiofilm activity of the different antibiotics was evaluated by monitoring for 48 h the heat produced by biofilm bacteria still viable on the beads (after treatment) re-inoculated in fresh medium. The obtained heat curves for each tested strain are displayed in Figure 4.

On the one hand, treatment with antibiotics alone showed a slight initial delay (up to 8 h) in heat production compared to the untreated sample (growth control), but overall, no remarkable antimicrobial effect against all tested EF strains, even at the highest antibiotic concentration tested (100× MIC), showing heat measurements after 48 h comparable to those of the growth control. Regarding the phage treatment, similarly to the effect observed with the antibiotics, the heat curves where alike to the curves from the growth control samples, confirming the outcomes observed with the time-killing assay, where biofilm re-growth could be seen after 24 h co-incubation.

To assess the antimicrobial effect of combining phage and antibiotic, a staggered exposure of *E. faecalis* biofilm, first to phages for 4 h followed by a 20-hour-exposure to different concentrations of antibiotic, was carried out. The incubation time with phage prior addition of antibiotics was based on the time-killing results (Figure 3).

In general, the combination of phage with vancomycin or gentamicin showed higher antibiofilm effects compared to the effect of each antimicrobial agent tested alone. Phage-antibiotic combinations revealed longer heat-suppression-times, lower heat production or even a complete absence of heat production during the whole monitoring time (48 h). Combining phage and gentamicin had a stronger antimicrobial effect than phage-vancomycin combinations against most *E. faecalis* strains, with the exception of EF 06, where we observed a similar outcome with both antibiotics in combination with phage. In case of EF 10, the combination of phage and any tested concentration of gentamicin, including the lowest 1× MIC, revealed a complete absence of heat production, correlating with an absence of bacterial regrowth for up to 48 h, mostly indicative of a significant reduction in the biofilm cell count or a complete biofilm eradication.

Another treatment approach investigated was the combination of phage with both antibiotics at different concentrations to assess potential positive or negative antibiotic interactions. Results mostly showed negative or no interaction for the combined antibiotic activity. We could observe, for EF 03, EF 10, and EF 14, that the combination of phage and gentamicin had a better outcome than when vancomycin was also included in the treatment. The addition of 10× MIC of vancomycin to phage-gentamicin (at 10× MIC) combination in the treatment against EF 06 had a better outcome than the single effect of phage-gentamicin or phage-vancomycin at the same concentrations.

### 3.5. Phage-Antibiotic Combinations against Vancomycin and Gentamicin High Level Resistant E. faecalis Biofilm

Daptomycin and fosfomycin were used to evaluate the antimicrobial activity of phage-antibiotic combinations against the biofilm of the vancomycin and high-level gentamicin resistant *E. faecalis* strain EF 04 by IMC.

A dose-dependent activity was observed when daptomycin was tested at different concentrations against EF 04, whereas fosfomycin showed a similar activity independently of the administered dose. Treatment with the phage resulted in a slight initial delay in heat production compared to the growth control (Figure 5).

The combination of phage CUB_EF04 and daptomycin revealed a considerable higher antimicrobial effect compared to the effect of phage or daptomycin alone. Daptomycin at 100× MIC combined with phage achieved the longest heat suppression (up to 34 h) and an overall lower heat production during the whole monitoring time (48 h) compared to the growth control. On the other hand, phage-fosfomycin combination showed a minor improvement in the antimicrobial effect relative to each antimicrobial alone, having an initial heat suppression time of up to 6 h compared to the growth control, but a higher overall heat production after 48 h.

When both antibiotics were combined with phage, no substantial changes in antimicrobial efficacy were observed. A slight longer suppression time and a lower overall heat production could be observed when combining phage with both antibiotics at 10× MIC compared to the combination of the phage and each antibiotic at the same concentration.

### 3.6. Assessment of Bacterial Resistance Development to Phage

After 48 h of co-incubation of bacterial strains EF 10 and EF 14 (parental strains) with phages CUB_EF10 and CUB_EF14 respectively, a bacterial colony (colony-variant) was recovered from each culture to evaluate the susceptibility to the exposed phage (CUB_EF10 in the case of EF 10 and CUB_EF14 in the case of EF 14) as well as to phages CUB_EF03, CUB_EF06 and CUB_EF10 or CUB_EF14, to which the parental strain had not been previously exposed. Top-Agar overlay spot assays revealed the development of resistance of the two tested *E. faecalis* strains (EF 10 and EF 14) to the phage to which they were exposed for 48 h (CUB_EF10 or CUB_EF14). A lysis zone could be observed on the agar plate when phage CUB_EF10 and CUB_EF14 were spotted on the parental strains (PS), indicating susceptibility to the phage, whereas no lysis was observed when spotting both phages on the colony-variants (CV), indicating emergence of resistance to the phage (Figure 6). We also observed that when CUB_EF10 phage was tested on the EF 14 PS, a turbid zone was visible, indicative of an absence or a low phage activity on this strain. In contrast to that, a clear lytic zone could be observed when the phage was spotted on the EF 14 CV. Apparently, the development of resistance in EF 14 to phage CUB_EF14 led to an increased susceptibility to the phage CUB_EF10.

No variations in susceptibility of all tested PS and CV strains to phages CUB_EF03 and CUB_EF06 were observed.

## 4. Discussion

The treatment of intraradicular infections is an important component of dental therapy. The reason for an irreversible pulpal inflammation is the excessive invasion of bacteria into the root canal system and the inability of the pulp to combat the invaders. The objective of a root canal treatment is to remove the pathogens (chemo-mechanical cleaning), shaping of the canal system, obturation, followed by coronal restauration. The outcome of this treatment depends directly on the pre-treatment status of the pulp and the periapical tissue. As previous studies have shown, teeth with a periapical radiolucency had the worst outcome with an apical healing up to 86% [33,34] and radiographic lesions were linked to biofilm formation in the root canal [35]. Hence, the presence of intraradicular biofilm and the bacterial persistence highly affects the treatment outcome. This fact reveals certain deficiencies of the gold standard of endodontic treatment and raises the question of whether new forms of biofilm eradication are necessary.

Bacteriophages seem to be a promising and innovative way in dentistry to fight resistant intraradicular pathogens, such as *E. faecalis*. As several studies have shown, phages reduce *E. faecalis* biofilm in dental ex vivo models and can be combined with common endodontic irrigants such as sodium hypochlorite and EDTA [24,36]. Furthermore, phage treatment against *E. faecalis* in an intraperitoneal and periapical rat infection model exhibited better outcomes (higher survival rate and normal anatomical diagnostic) compared to antibiotic treatment, as shown by Xiang et al. in a recent study [37]. These results represent a very promising alternative to conventional root canal treatment, especially for retreatments or teeth with an apical radiolucency.

As shown in our study, sewage water from cesspools is an effective and inexpensive source to isolate bacteriophages against *E. faecalis* to target antibiotic resistant strains and biofilm formation. Despite the high specificity that phages often exhibit, infecting individual bacterial species or even individual bacterial strains, our host range analysis revealed a broad coverage of the number of *E. faecalis* strains susceptible to at least one of the isolated phages, with 5 strains out of 16 not susceptible to any phage.

For our study, we focused on investigating the antibiofilm activity of five different phages towards the host-bacterial strain used during the isolation step. To this end, we used an optimized in vitro biofilm model using porous glass beads as a surface to grow biofilm [38], with an obtained cell count of 10^8^ CFU/mL on average, indicative of a strong mature biofilm. Time-killing assays showed a significant biofilm reduction in all tested strains during the initial incubation time (up to 6 h) before bacterial re-growth could be observed after 24 h co-incubation, possibly due to the development of bacterial resistance to phage leading to a repopulation of the biofilm.

Indeed, phage resistance has also been reported in phage therapy in humans [39] representing a potential barrier to the implementation of phage therapy. In that sense, our study also provides valuable data on the susceptibility of phage-resistant mutants to different phages. For instance, we observed that emergence of bacterial resistant after co-incubation with one phage could lead to a higher susceptibility of the resistant strain to another phage for which the original strain showed weak susceptibility. Further research could provide more insights into the mechanisms involved in the observed correlation between resistance to one phage and increased susceptibility to another. Some studies have found that *E. faecalis* phage resistance involves the mutation of phage receptors, including membrane protein PIP [40] or the enterococcal polysaccharide antigen (Epa) [41].

Combination of phages and antibiotics is a promising strategy to reduce the dose of antibiotics and the development of antibiotic resistance during treatment. As shown in previous studies [42,43,44] a staggered administration, adding first the phage and then the antibiotic, resulted in better outcomes than a simultaneous exposure to both antimicrobials. Thus, we used a staggered approach in our study. By the use of isothermal microcalorimetry, we were able to systematically assess the antimicrobial activity of different treatment conditions against biofilms. The combined effect of phages and antibiotics in the treatment of *E. faecalis* biofilms proved to be more efficient than each antimicrobial alone. In particular, the combination of phage and gentamicin revealed a higher efficacy compared to the combination with vancomycin. This may be due to the higher antibiofilm activity shown by gentamicin compared to vancomycin against *E. faecalis*, also seen in other in vitro and in vivo studies [45,46].

The spread of vancomycin-resistant enterococci (VRE) is increasing, becoming a major clinical problem due to the intrinsic resistance of these strains to most commonly used antibiotics and thus limiting therapeutic options [47,48]. Our study showed the successful application of phage in combination with daptomycin against the VRE strain, achieving a complete inhibition of the biofilm at an antibiotic concentration of 0.8 g/L, an achievable dose by local antibiotic administration in clinical practice.

Despite our efforts to bring further insights in the use of phage therapy as potential treatment approach for post-treatment apical periodontitis, when trying to draw conclusions or make clinical extrapolations, it is important to consider some limitations of our study. Compared to the complexity and individuality of intraradicular biofilms in post-treatment apical periodontitis, mainly of polymicrobial nature [49]. Our study investigated mono-species biofilms using individual *E. faecalis* strains, where only combinations of one phage with its host bacterial strain were tested. Additional studies on the activity of single or combined phages across different bacterial isolates might bring further insights on the therapeutic potential of phages. Furthermore, it is a matter of debate whether *E. faecalis* is the main pathogen associated with endodontic treatment failure. Recent studies reveal that *Parvimonas micra* and *Fusobacterium nucleatum* are more prevalent in previously treated teeth with a persistent infection and may play a more important role in the process of apical inflammation. Nevertheless, in most cases *E. faecalis* appears to be present in root canal treated teeth with an apical lesion, being able to withstand common endodontic treatment options [50,51,52]. Regarding the potential use of phages in endodontics, in any case an individual treatment approach will be necessary, in which phages active against the pathogenic strain causing the infection should be used.

In summary, phage therapy holds a very high potential to eradicate persistent intraradicular *E. faecalis* biofilm and may present a rewarding addition to the common endodontic treatment.

## Figures and Tables

**Figure 1 pharmaceutics-14-01779-f001:**
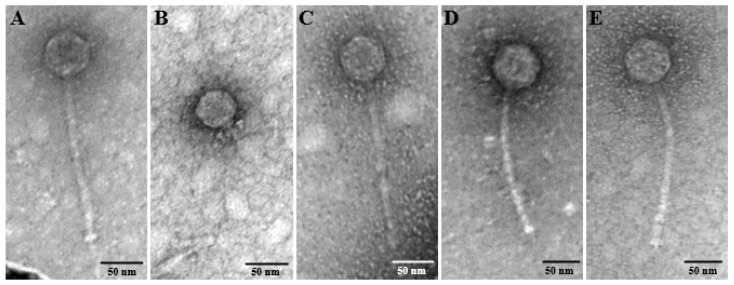
TEM image of the bacteriophage’s virions (**A**) CUB_EF03, (**B**) CUB_EF04, (**C**) CUB_EF06, (**D**) CUB_EF10, and (**E**) CUB_EF14.

**Figure 2 pharmaceutics-14-01779-f002:**
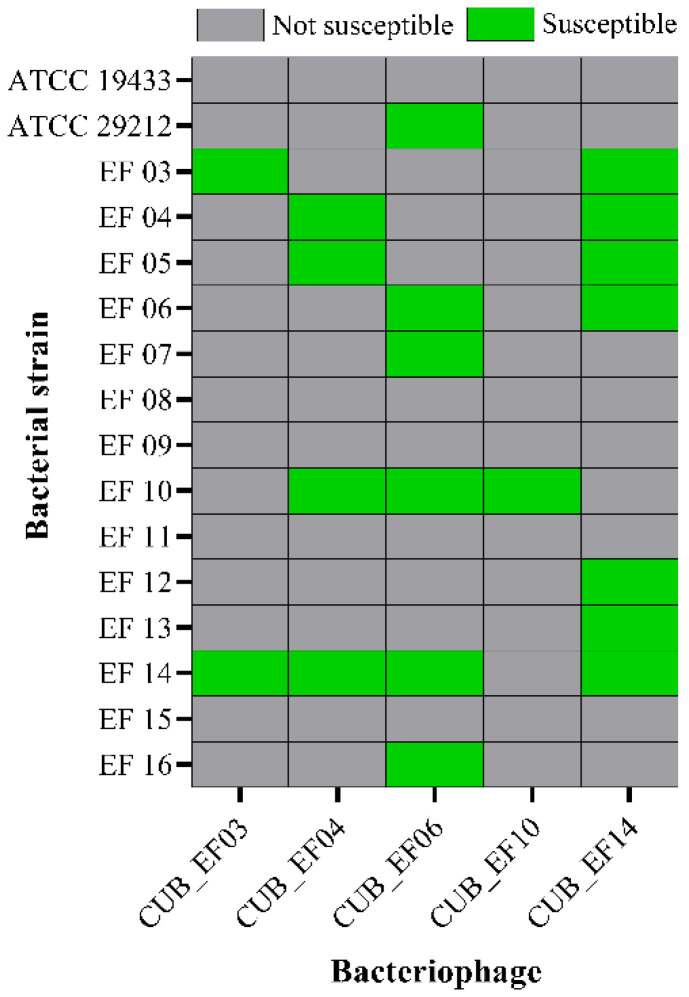
Host range of isolated bacteriophages among *E. faecalis* strains.

**Figure 3 pharmaceutics-14-01779-f003:**
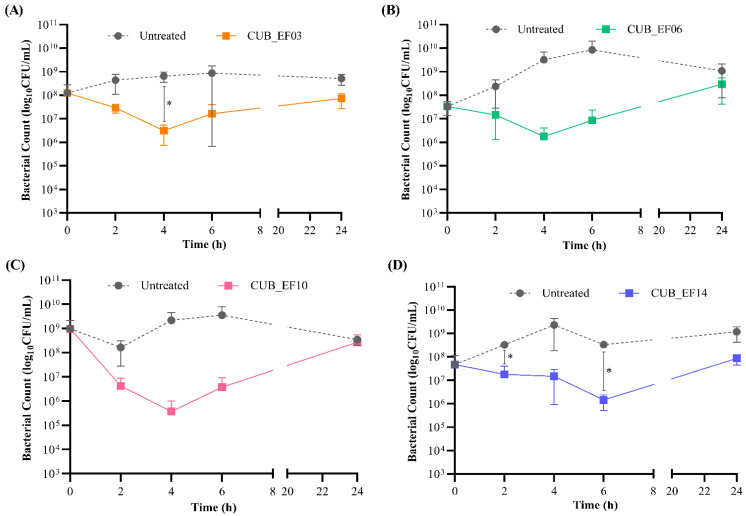
Time-killing curve of *E. faecalis* biofilms from host strains (**A**) EF 03, (**B**) EF 06, (**C**) EF 10, and (**D**) EF 14 treated with the respective phage (10^8^ PFU/mL) and untreated monitored at 2 h intervals for the first 6 h and after 24 h. Data are expressed as mean ± SD. Results were statistically analyzed using a multiple *t* test analysis integrated in GraphPad Prism 6; *p*-values < 0.05 were considered significant (*).

**Figure 4 pharmaceutics-14-01779-f004:**
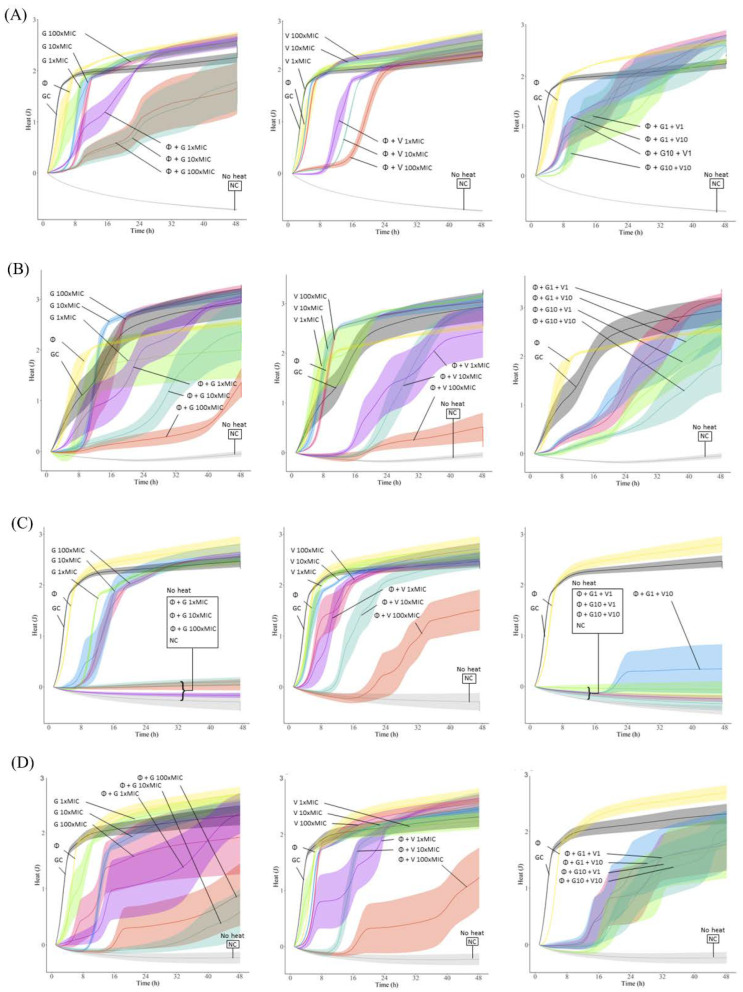
Heat (J) curves of *E. faecalis* (**A**) EF 03, (**B**) EF 06, (**C**) EF 10 and (**D**) EF 14 biofilms exposed to phage (Φ at 10^8^ PFU/mL), gentamicin (G), or vancomycin (V) at different concentrations (1×, 10× or 100× MIC), or with staggered phage-antibiotic combinations (Φ + antibiotic). Each curve shows the heat produced over time by viable bacteria in the biofilm after 24 h of treatment. GC represents the growth control sample not exposed to any antimicrobials. NC represents de negative control for media sterility. Data are expressed as mean ± SE. Plots were prepared using RStudio [30]. Solid lines represent the mean and corresponding shaded regions the standard error.

**Figure 5 pharmaceutics-14-01779-f005:**
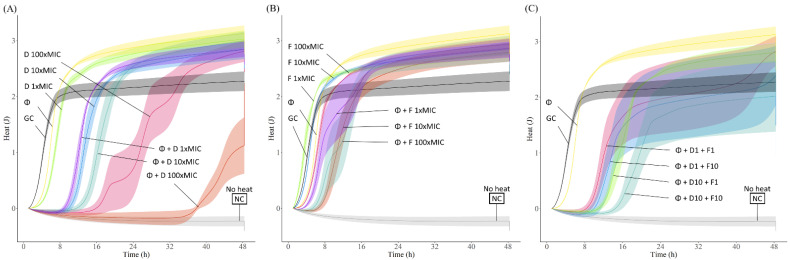
Heat (J) curves of EF 04 biofilm exposed to (**A**) phage CUB_EF04 (Φ at 10^8^ PFU/mL) plus daptomycin (D) at different concentrations (1×, 10× or 100× MIC), (**B**) phage CUB_EF04 (Φ at 10^8^ PFU/mL) plus fosfomycin (F) at different concentrations (1×, 10× or 100× MIC), or (**C**) phage CUB_EF04 (Φ at 10^8^ PFU/mL) plus both antibiotics at different concentrations (1×, 10× MIC). Each curve shows the heat produced by viable bacteria in the biofilm after 24 h of treatment. GC represents the growth control sample not exposed to any antimicrobials. NC represents de negative control for media sterility. Data are expressed as mean ± SE. Plots were prepared using RStudio [30]. Solid lines represent the mean and corresponding shaded regions the standard error.

**Figure 6 pharmaceutics-14-01779-f006:**
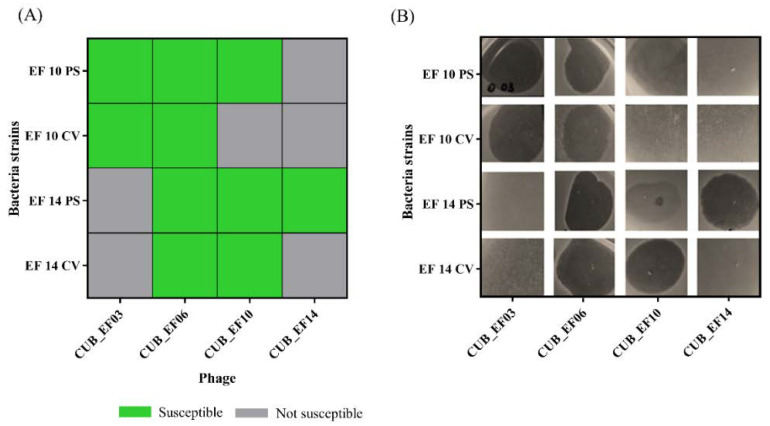
(**A**) Heat-map showing susceptibility of the parental (PS) and the colony-variant (CV) *E. faecalis* strains EF 10 and EF 14 to phages CUB_EF03, CUB_EF06, CUB_EF10 and CUB_EF14. (**B**) Representative images of Top-Agar overlay spot tests.

**Table 1 pharmaceutics-14-01779-t001:** Minimal inhibitory concentration (MIC) values of planktonic *E. faecalis* (EF) strains determined by broth microdilution.

Antibiotic	MIC EF 03	MIC EF 06	MIC EF 10	MIC EF 14	MIC EF 04
Vancomycin	2 µg/mL	2 µg/mL	2 µg/mL	2 µg/mL	-
Gentamicin	64 µg/mL	64 µg/mL	128 µg/mL	64 µg/mL	-
Fosfomycin	-	-	-	-	32 µg/mL
Daptomycin	-	-	-	-	8 µg/mL

## Data Availability

Not applicable.

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
