# Peer review of "In Vitro Evaluation of Five Newly Isolated Bacteriophages against E. faecalis Biofilm for Their Potential Use against Post-Treatment Apical Periodontitis"

_pharmaceutics, 2022, doi:10.3390/pharmaceutics14091779_

Round 1

Reviewer 1 Report

In the article "In vitro evaluation of five newly isolated bacteriophages against E. faecalis biofilm for their potential use against post-treatment apical periodontitis", the authors characterize the properties of five new phages including their host range and bacteriacidal capacity. The experiments are well thought out, and compelling. I have a few points I would like to see clarified prior to publication. 

1. Host range experiments demonstrated that most of the phages are capable of infecting a range of E. faecalis strains. Further experiments showed use of only one host strain with its similarly named phage. Would you please include a brief description in the results or discussion section as to why these particular pairings were chosen rather than attempting to examine the properties of each phage across all of its potential hosts?

2. Similarly, in the section looking at the antibiotic resistant strain EF04, please explain why use of CUB_EF14 was not also examined.

3. While the experimental design for 3.6 "Assessment of bacterial resistance to development to phage" was described in the materials and methods, it is not clear in the text of the results section that exposure to CUB_EF03, CUB_EF06, CUB_EF10, and CUB_EF14 is following a previous exposure to either strain's paired phage. As this is essential to the significance of the results presented in Figure 6, it would be nice to have this clarified here as well. 

Reviewer 2 Report

Dear Authors, 

I read the manuscript with interest. I think the research is well planned and carried out, and the manuscript is well written. I just have a few comments:

1. You reported that five different phages were isolated (CUB-EF03, CUB-04, CUB-EF06, CUB-EF10, CUB-EF14, but in the subsection 3.3. Time-Killing Assay results were described only for four phages (CUB-EF03, CUB-FE06, CUB-10, CUB-14). It should be explained why only these four phages are selected for the assay. 

2. Similar to the above, the Assessment of bacterial resistance development to phage (subsection 3.6.) does not present results for CUB_EF04. Please, explain this. 

In the 3.6 subsection, the results are described for CUB-EF10, CUB-EF14, CUB-EF03, and CUB-EF06. In the corresponding subsection 2.9 Assessement of Bacterial Resistance Development to Phage in the Results paragraph, only CUB-EF10 and CUB-EF14 are mentioned. Why?
